# Reference Values for Fitness Level and Gross Motor Skills of 4–6-Year-Old Chilean Children

**DOI:** 10.3390/ijerph17030797

**Published:** 2020-01-28

**Authors:** Andrés Godoy-Cumillaf, José Bruneau-Chávez, Paola Fuentes-Merino, Jaime Vásquez-Gómez, Mairena Sánchez-López, Celia Alvárez-Bueno, Iván Cavero-Redondo

**Affiliations:** 1Grupo de Investigación en Educación Física, Salud y Calidad de Vida, Facultad de Educación, Universidad Autónoma de Chile, 4810101 Temuco, Chile; andres.godoy@uautonoma.cl (A.G.-C.); jose.bruneau@uautonoma.cl (J.B.-C.); paola.fuentes@uautonoma.cl (P.F.-M.); 2Vicerrectoría de Investigación y Postgrado, Centro de Investigación en Estudios Avanzados del Maule (CIEAM), Universidad Católica del Maule, 3460000 Talca, Chile; jvasquez@ucm.cl; 3Health and Social Research Center, Universidad de Castilla-La Mancha, 16071 Cuenca, Spain; mairena.sanchez@uclm.es (M.S.-L.); ivan.cavero@uclm.es (I.C.-R.); 4Faculty of Education, Universidad de Castilla-La Mancha, 13071 Ciudad Real, Spain

**Keywords:** motor competence, strength, speed/agility, cardiorespiratory fitness, flexibility, balance, aiming, catching

## Abstract

In childhood, fitness level is considered an important indicator of health, while gross motor skills are the basis of future motor competence. So far, no reference values have been found for the Chilean population. Therefore, this study aims to provide fitness level and gross motor skill reference values by gender and age of Chilean children aged 4–6 years. A cross-sectional analysis was conducted that included 728 children between 4 and 6 years old from the La Araucanía region of Chile. To assess the fitness level, the 20-m shuttle run test, standing long jump, handgrip dynamometry, 4x10m shuttle run, and Sit and Reach tests were used. Gross motor skills were assessed by five tests including aiming and catching and balance motor tasks. For fitness level, boys have better values in the long jump and dynamometry test, while girls have better values in flexibility. For estimated maximum volume of oxygen, at 5 years old there are significant differences in favour of boys, while at 6 years old in favour of girls. No statistically significant differences in speed/agility by gender were found. For gross motor skills, boys obtain higher values for catching and aiming tests, and girls for balance. The reference values for fitness level and gross motor skills shown in this study could aid physical education and health professionals in identifying children with low reference values, as well as in establishing objectives that will help to improve their health.

## 1. Introduction 

High fitness level is considered an important indicator of health in childhood [1,2], while low fitness level is associated with an increased risk of acquiring cardiovascular diseases in adulthood [3,4,5]. Moreover, cardiorespiratory fitness is a basic component of a healthy lifestyle [6,7], its optimum levels being associated with improved cardiovascular, skeletal, and mental health [1,2]. It should be highlighted that both fitness level and cardiorespiratory fitness depend on genetic factors [8], including anatomical and physiological, and environmental factors [9], such as behaviour and lifestyle aspects. For muscular strength from childhood to adolescence, the increased levels are associated with a decrease in total adiposity [1] and lower metabolic risk [10,11,12]. Likewise, speed/agility improvements are related to higher bone mineral density and accumulation of bone mass in later stages of life [4,5]. Also, flexibility plays an important role in the range and coordination of movements, this ability being associated with good results in terms of fitness level [13]. Considering everything mentioned above, previous studies [14] have suggested that it is necessary to include tests to assess the fitness level of children in the school settings, taking even more into account that Chile lacks records on indicators of fitness level [15], presenting a considerable gap in the literature for children 12 years old or younger [16]. 

Motor competence, defined as the degree of performance in a range of motor tasks that are rated on good control and movement coordination [17], is important because of its observed implications in the physical, mental, and social development of children and adolescents [18,19]. Motor competence is composed of fine and gross motor skills, with the former being important in the academic and social fields [18], and the latter developing the basis of future motor competence. Therefore [20], if children lack mastery in these skills, it is probable that they will have limited opportunities to successfully participate in the different physical activities throughout life [21], and as a result, delays or deficiencies in gross motor skills may affect children and adults’ fitness levels [22].

In considering early prevention of risk factors for noncommunicable diseases (obesity, diabetes, cardiovascular disease, and others), assessment of fitness levels and gross motor skills is necessary for decision-making and intervention aimed at promoting healthy behaviour. These assessments require the existence of updated reference values that allow the categorization of individuals and groups according to fitness levels and gross motor skills. Additionally, these values might be established by sex as previous evidence has demonstrated significant differences in fitness levels and gross motor skills between boys and girls at early ages [23,24,25,26]. 

For 4–6-year-old children, there are international established reference values, for both fitness level [14,27,28] and gross motor skills [22,29]. However, until now there is no clear cut-off for the Latin American population, specifically Chile, which makes it necessary to have their own values due to sociodemographic factors, the environment, and adequate nutrition of this population, which lead them to have different characteristics. Therefore, the present study aims to describe fitness levels and gross motor skills by gender and age, and to provide the first reference values in children aged 4–6 years, from La Araucanía Region, Chile. 

## 2. Methods

This is a cross-sectional analysis of data from baseline measurements of a randomized controlled trial (RCT) registered in ClinicalTrial.gov (NCT04194580), aimed at assessing the effectiveness of a physical activity intervention in preventing and treating obesity and excess weight in children aged 4–6 years. The RCT study included 836 schoolchildren from the city of Pitrufquén, La Araucanía region, Chile. For the current study, data from 728 (87%) children who had valid fitness level and gross motor skill measurements were used (Figure 1). The exclusion criteria were: (i) children not having legal guardian consent or not having the child’s consent to carry out evaluations; (ii) children having some type of physical and/or mental disorder; and (iii) children suffering a chronic illness that could prevent participation in the physical activity intervention. The study was approved by the Scientific Ethical Committee of the Universidad Autónoma de Chile (Nº11–19). The anthropometric, fitness level, and gross motor skill measurements were carried out by physical activity science graduates trained for this study to guarantee standardization. These graduates were responsible for ensuring that the instructions and procedures were understood in each test. 

### 2.1. Fitness Level

Cardiorespiratory fitness was assessed using the 20-m shuttle run test, which provides valid and reliable information on maximum aerobic capacity in children [30]. Estimated maximum oxygen volume value (VO2max) was calculated by using the preschool-adapted 20-m shuttle-run (PREFIT) formula [31]. The children were encouraged to run as long as possible during the test, and the last half of the stage reached by the participants was recorded. Muscle strength was assessed using the standing long jump test for lower limb strength; the children were asked to jump horizontally to reach the maximum distance and the result was recorded in cm as the best value of the three attempts made. For upper limb strength, handgrip dynamometry (Takei 5401) was used, two attempts were made for each hand, the highest values were recorded in kg. Speed/agility was measured using the 4 × 10 m shuttle run, two attempts were made with a 5-min rest between them and the lowest value of the two was recorded in seconds. Flexibility was measured with the Sit and Reach test, the result was recorded in cm as the best value of the two attempts made. These evaluation procedures are part of batteries used to measure fitness level in children [32,33], and provide valid and reliable information [30,32,34,35]. The brief definition of each test is given in Appendix A.

### 2.2. Gross Motor Skills

The validated Spanish version of the second edition of the Movement Assessment Battery for Children (MABC-2) [36], which has proven to be an instrument with adequate psychometric properties of reliability (α de Cronbach > 0.60; *κ* = 1; CCI = 0.85–0.99) [37], was used to measure gross motor skills. This tool was developed to be used in clinical and educational settings. The version with the 4–6 years age range was used. The motor tasks used to assess gross motor skills were: catching and aiming (2 tests: catching a beanbag and throwing a beanbag onto a mat), and static and dynamic balance (3 tests: balance for each leg—static balance; tip-toe walking and jumping on a mat—dynamic balance). 

### 2.3. Data Analysis

The adjustment to normal distribution of the different variables was evaluated both by graphs and by the Kolmogorov-Smirnov test. Fitness levels and gross motor skills by age (years) and gender are presented with their mean and standard deviations. Since all variables had a normal distribution, we used parametric tests in the analysis. 

Gender differences in fitness levels and gross motor skills were tested using the Student’s t statistic (for independent samples). 

In order to establish the influence of gender, age in months, height, weight, and body mass index (BMI) on fitness level and gross motor skills, a linear regression analysis was performed adding these variables as fixed factors [38,39,40]. 

The ANCOVA model was used to assess differences in fitness level and gross motor skills, controlling for age in months and BMI, by gender.

The percentile values (P10, P20, P30, P40, P50, P60, P70, P80, P90, and P100) were calculated by age and gender for each fitness level and gross motor skills test. Statistical significance was assumed for *p* ≤ 0.05. The IBM SPSS Statistics version 25 was used for the data analysis. 

## 3. Results

The final sample for this analysis included 728 children (332 boys and 396 girls). The mean weight, height, and BMI values for each age (4, 5, and 6 years old) and gender ranged from 20.0 to 27.1 kg, from 106.5 to 120.3 cm, and from 17.1 to 19.1 kg/m^2^, respectively (Table 1).

For fitness level, for unadjusted and adjusted values for age and BMI, boys showed better values than girls in the jump (*p* from <0.00 to 0.00), right and left dynamometry (*p* from <0.00 to 0.12), dominant hand dynamometry (*p* from <0.00 to 0.00), no dominant hand dynamometry (*p* from <0.00 to 0.00), and speed/agility tests (*p* from 0.01 to 0.72), being all statistically significant except left dynamometry at 5 years and speed/agility at 5 and 6 years. Moreover, girls showed significantly higher values than boys in flexibility at all ages (*p* from <0.00 to 0.00). Finally, differences in estimated VO2max were significant for boys at 5 years (*p* = 0.00) and for girls at 6 years (*p* = 0.04) (Table 2).

Regarding gross motor skills, for unadjusted and adjusted values for age and BMI, boys obtained better values than girls for the catching and aiming tests, showing statistically significant differences for the catching at 4 (*p* = 0.02) and 5 (*p* = 0.04) years and for the aiming at 5 years (*p* = 0.00). For both static and dynamic balance, girls obtained higher values than boys (*p* from <0.00 to 0.92), being all statistically significant except right balance at 4 years, left balance at 4 and 5 years and tip-toes at 5 years (Table 3).

Age in months, height, weight, and BMI were strongly associated with fitness level and gross motor skills in both boys and girls (Table 4).

Finally, Table 5 and Table 6 (girls), and Table 7 and Table 8 (boys) show percentile values for each physical fitness and gross motor skills test. Fitness level and gross motor skills were improved with age.

## 4. Discussion

This is the first study to describe fitness levels and gross motor skills in children from 4 to 6 years old, from La Araucanía, Chile. In addition to providing percentile values by age and gender, the results will complement existing information for health parameters such as BMI, waist circumference, weight, and height [41]. 

The results of this study show that, regarding fitness level, boys obtained better results in tests of strength, while girls obtained better results in flexibility. These results were consistent with those reported in similar studies [23,24], and are largely explained by the physical differences in muscle mass and height between genders [28]. For estimated VO2max, there were no significant differences by gender at 4 years, whereas at 5 years there were significant differences in favour of boys, and at 6 years in favour of girls. Other previous studies did not report these differences in favour of girls [27,35,42,43,44], which could be because the maximum values of VO2max are relative (mL/kg/min) and these differences could be explained by weight; however, at 6 years old the differences are minimal (0.1 kg), which could explain that the value in favour of girls is a consequence of increased physical activity that would benefit an increase in VO2max.

Regarding flexibility, there was an improvement from 4 to 5 years, but a decrease at 6 years, which would be in line with findings from a research carried out with children older than those evaluated in this study [13,27], where there was a progressive decline in this ability. However, this decrease may not only be explained by age, but also by scarce or no training carried out in this ability, since the training of flexibility in a specific manner will lead to its maintenance or improvement, promoting coordination and range of movements [13,19].

For gross motor skills, boys had better values in the catching and aiming tests, while girls obtained better results for static and dynamic balance, which is consistent with findings from other studies that used similar tests [27,43,44]. 

In general, at a higher age, higher values for both genders were clear, and differences in favour of boys started to increase, which is in line with other studies [22,24,27,28]. These differences between genders, both for fitness level and gross motor skills, might not be explained by biological factors [8] alone, but also by socialization processes, where the type, duration, and frequency of the physical activities performed are different for the two genders [45], which is supported by Bandura [46], who states that school-age children are stereotyped in certain roles based on their gender. This is supported by noticing that it is common to see boys playing games or exercises with a ball while girls carry out activities and games with an expressive-rhythmic-motor component [47,48]. However, currently we should now consider whether these stereotypes are changing, and whether this may or may not be related to the significant differences in favour of girls at 6 years, which were found in relation to estimated VO2max. 

Therefore, the results of this research will constitute an instrument that can be added to those that already exist for detecting and monitoring the health of children; providing an adequate basis for carrying out physical activity performance interventions, especially those aimed at the development of fitness levels and gross motor skills; and helping to prevent and treat diseases that currently affect the health of the population. In this sense, the schools play an important role since they are institutions where children spend a great deal of their time. Also, it has been demonstrated through meta-analyses [49] that various promotive strategies for physical activity in schools have been effective, and that these physical activity interventions should be promoted by educational institutions, combining with families and government agencies entrusted with public health. Thus, it is recommended that from an early age, girls and boys are encouraged to perform in different types of physical activity where there is no sex-based exclusion, which favours a similar development in physical condition and motor competence. 

The potential limitations of this study are: (i) its transversal nature, in the absence of other variables that could give a greater understanding of the phenomenon and establish cause and effect relationships; (ii) the sample is not probabilistic and comes from only one region of Chile, so inferences to Chilean children should be made with caution. Therefore, more studies are needed at the national level that provide data to verify what is found here. However, the inclusion of poorly or barely studied ages could be a good first approach that may be useful for other research or health interventions.

It would be advisable to carry out further studies to deepen the research on this topic, incorporating variables such as diet, physical activity during free time, as well as the consideration of stereotypes (that change over time), and culture.

## 5. Conclusions

Our findings show that for fitness level and gross motor skills, boys have better values than girls in strength, catching and aiming tests, while girls have better results in flexibility and balance. Additionally, depending on age, the estimated VO2max shows significant differences in favour of boys or girls. No statistically significant differences in speed/agility are shown by gender. The study provides specific reference values for children aged 4 to 6 years by age and gender, which will aid physical education and health professionals in identifying those with low physical fitness and gross motor skills values, in order to establish goals to help improve their health, whereas for those with high values in abilities considered important for sports, this will help in early detection of higher competence.

## Figures and Tables

**Figure 1 ijerph-17-00797-f001:**
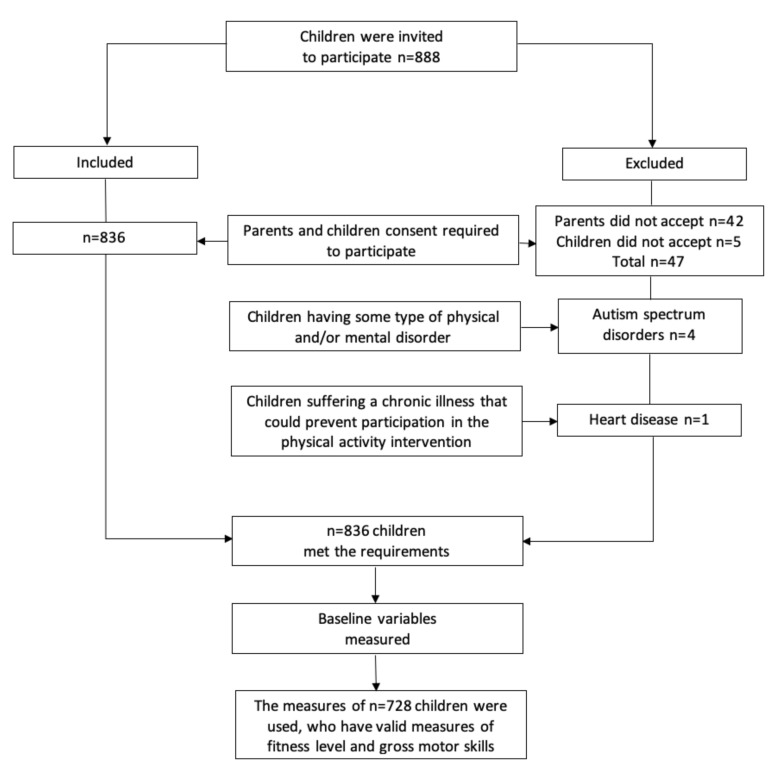
Flow diagram of participant recruitment and exclusion with reasons.

**Table 1 ijerph-17-00797-t001:** Anthropometric characteristics (Mean ± SD), by age and gender.

Age (years)	Weight (kg)	Height (cm)	BMI (kg/m^2^)
**Boys (332)**			
4 (54)	20.0 ± 3.1	107.7 ± 2.7	17.1 ± 2.1
5 (112)	23.6 ± 4.3	114.5 ± 4.5	17.9 ± 2.4
6 (166)	27.0 ± 5.3	120.3 ± 5.6	18.5 ± 2.6
**Girls (396)**			
4 (92)	20.0 ± 3	106.5 ± 5.2	17.5 ± 1.5
5 (126)	22.5 ± 3.8	111.6 ± 4.7	18.0 ± 2.4
6 (178)	27.1 ± 5.5	118.8 ± 5.7	19.1 ± 2.9

BMI, body mass index (calculated as weight in kg divided by height in m^2^).

**Table 2 ijerph-17-00797-t002:** Fitness level values (Mean ± SD), by age and gender.

Fitness Level				
		Boys (*n* = 332)	Girls (*n* = 396)	*p*
**Lower limb strength**				
Long jump (cm)				
4	Unadjusted	76.6 ± 23.5	65.7 ± 18.9	**0.00**
	Adjusted	76.7 ± 23.5	65.2 ± 18.7	**0.00**
5	Unadjusted	88.9 ± 21.5	79.4 ± 14.7	**0.00**
	Adjusted	88.6 ± 21	79.9 ± 14.5	**0.00**
6	Unadjusted	96.5 ± 24.4	86.2 ± 18.3	**0.00**
	Adjusted	96.1 ± 24	86 ± 14.7	**0.00**
**Upper limb strength**				
Right Dynam (kg)				
4	Unadjusted	6.1 ± 1.6	5.2 ± 2.1	**0.00**
	Adjusted	6 ± 1.4	5.3 ± 2	**0.00**
5	Unadjusted	7.9 ± 1.6	6.7 ± 1.3	**0.00**
	Adjusted	8.1 ± 1.7	6.9 ± 1.1	**0.00**
6	Unadjusted	9 ± 2.1	8.1 ± 2.1	**0.00**
	Adjusted	9.1 ± 2.2	8.4 ± 2.2	**0.01**
Left Dynam (kg)				
4	Unadjusted	6.1 ± 1.5	5.1 ± 2.1	**0.00**
	Adjusted	6.3 ± 1.7	5.3 ± 2.1	**0.00**
5	Unadjusted	7.1 ± 1.9	6.7 ± 1.7	0.12
	Adjusted	7.2 ± 1.7	6.9 ± 1.5	0.14
6	Unadjusted	9.2 ± 2.2	7.9 ± 2	**0.00**
	Adjusted	9.5 ± 2	8.1 ± 1.7	**0.01**
Dominant hand (kg)				
4	Unadjusted	6.3 ± 1.4	5.3 ± 2	**0.00**
	Adjusted	6.5 ± 1.7	5.7 ± 2.3	**0.00**
5	Unadjusted	7.9 ± 1.4	6.9 ± 1.8	**0.00**
	Adjusted	8.1 ± 1.6	7.1 ± 2	**0.00**
6	Unadjusted	9.2 ± 1.4	8.2 ± 2.1	**0.00**
	Adjusted	9.8 ± 1.5	9.1 ± 3.2	**0.00**
No dominant hand (kg)				
4	Unadjusted	5.4 ± 1.3	4.7 ± 1.8	**0.00**
	Adjusted	5.4 ± 1.5	4.9 ± 1.6	**0.00**
5	Unadjusted	6.8 ± 1.6	6.4 ± 1.6	**0.00**
	Adjusted	7.1 ± 1.9	6.9 ± 2	**0.00**
6	Unadjusted	8.5 ± 1.2	7.1 ± 1.7	**0.00**
	Adjusted	9 ± 1.6	7.7 ± 1.1	**0.00**
**Flexibility**				
Sit and reach (cm)				
4	Unadjusted	27.2 ± 3.4	29.1 ± 4.1	**0.00**
	Adjusted	27.4 ± 3.5	29.5 ± 4	**0.00**
5	Unadjusted	31.1 ± 4	32.6 ± 4.6	**0.00**
	Adjusted	31.4 ± 4.2	33.1 ± 4	**0.00**
6	Unadjusted	29.7 ± 5.4	32.4 ± 5.2	**0.00**
	Adjusted	29.9 ± 5.2	33 ± 4.9	**0.00**
**Speed/Agility ^¥^**				
4 × 10 (sec)				
4	Unadjusted	17.4 ± 1.5	18.2 ± 1.8	**0.01**
	Adjusted	17.3 ± 1.2	18 ± 1.5	**0.00**
5	Unadjusted	16.8 ± 2.4	17.4 ± 2.1	0.06
	Adjusted	16.5 ± 2	17.2 ± 2	0.06
6	Unadjusted	16.3 ± 1.8	16.3 ± 1.3	0.72
	Adjusted	15.9 ± 2	16 ± 1.8	0.72
**Cardiorespiratory fitness**				
Course navette (stage) ^‡^				
4	Unadjusted	1.2 ± 0.6	1.3 ± 0.6	0.59
	Adjusted	1.1 ± 0.8	1.1 ± 0.4	0.59
5	Unadjusted	1.7 ± 0.6	1.4 ± 0.6	**0.00**
	Adjusted	1.8 ± 0.6	1.6 ± 0.6	**0.00**
6	Unadjusted	1.8 ± 0.9	2 ± 0.8	0.10
	Adjusted	2 ± 0.6	2.2 ± 0.8	0.10
VO2max (mL/kg/min)				
4	Unadjusted	47.1 ± 1.3	47 ± 1.4	0.66
	Adjusted	47.5 ± 1.5	47.6 ± 1.8	0.66
5	Unadjusted	47.8 ± 1.4	47.2 ± 1.3	**0.00**
	Adjusted	47.5 ± 1.6	47 ± 1.1	**0.00**
6	Unadjusted	48 ± 2	48.4 ± 1.8	0.04
	Adjusted	48.1 ± 2.2	48.3 ± 1.9	0.04

**^¥^** Less time (in sec) indicates better fitness level, ^‡^ 1 stage = 1 min. The values in bold indicate a statistical significance for *p* < 0.05.

**Table 3 ijerph-17-00797-t003:** Gross motor skills values (Mean ± SD), by age and gender.

Gross Motor Skills				
		Boys (*n* = 332)	Girls (*n* = 396)	*p*
**Catching and aiming**				
Catching (number)				
4	Unadjusted	7.1 ± 2.7	6 ± 2.9	**0.02**
	Adjusted	7 ± 2.5	5.8 ± 2.4	**0.03**
5	Unadjusted	8.5 ± 2.3	7.8 ± 2.4	**0.04**
	Adjusted	8.6 ± 2.5	7.9 ± 2.2	**0.02**
6	Unadjusted	8.9 ± 2	8.9 ± 1.7	0.87
	Adjusted	9.1 ± 1.6	9 ± 2.1	0.85
Aiming (number)				
4	Unadjusted	3.8 ± 2.6	4.1 ± 2.6	0.49
	Adjusted	3.7 ± 2.4	4.2 ± 2.2	0.45
5	Unadjusted	4.5 ± 2.1	3.7 ± 2.3	**0.00**
	Adjusted	4.7 ± 2.3	3.8 ± 2	**0.00**
6	Unadjusted	5.2 ± 2.2	5.1 ± 3.4	0.73
	Adjusted	5.4 ± 2.5	5.3 ± 3.8	0.69
**Balance**				
Right balance (sec)				
4	Unadjusted	10.3 ± 6,5	15,6 ± 10.3	**0.00**
	Adjusted	10.1 ± 5.9	16 ± 9.9	**0.00**
5	Unadjusted	19 ± 10	16.7 ± 9.2	0.06
	Adjusted	18.6 ± 10	17.1 ± 9	0.06
6	Unadjusted	18.5 ± 10.4	24,9 ± 8.1	**0.00**
	Adjusted	18.9 ± 10	25 ± 7	**0.01**
Left balance (sec)				
4	Unadjusted	12.7 ± 9.7	12.8 ± 9	0.92
	Adjusted	12.9 ± 9.3	13 ± 8.9	0.88
5	Unadjusted	17.2 ± 10.7	19.1 ± 9.8	0.16
	Adjusted	17.5 ± 9.6	19.2 ± 9	0.19
6	Unadjusted	20.2 ± 10	23.8 ± 8.5	**0.00**
	Adjusted	20.9 ± 10	24.1 ± 8	**0.00**
Dominant leg (sec)				
4	Unadjusted	10.4 ± 5.8	16.6 ± 9	**0.00**
	Adjusted	10.2 ± 5.9	16.7 ± 8.1	**0.00**
5	Unadjusted	19.3 ± 9.4	17.1 ± 8	**0.00**
	Adjusted	19.5 ± 9	17.6 ± 8.2	**0.00**
6	Unadjusted	19.4 ± 9.9	25.9 ± 7.1	**0.00**
	Adjusted	21.6 ± 9.1	26 ± 7	**0.00**
No dominant leg (sec)				
4	Unadjusted	11.5 ± 9	12.6 ± 8.8	**0.00**
	Adjusted	11.4 ± 8.8	12.7 ± 8.7	**0.01**
5	Unadjusted	17.1 ± 9.2	18.8 ± 9	**0.00**
	Adjusted	17.3 ± 9.6	18.9 ± 9.1	**0.00**
6	Unadjusted	17.2 ± 8.8	24 ± 8.2	**0.00**
	Adjusted	17.6 ± 8.1	24.2 ± 8.3	**0.00**
Tip-toes (steps)				
4	Unadjusted	11.3 ± 5.4	13.5 ± 3.2	**0.00**
	Adjusted	11.1 ± 5	13.1 ± 3	**0.00**
5	Unadjusted	13.1 ± 3.9	13.7 ± 3.3	0.23
	Adjusted	13.5 ± 4.1	13.9 ± 2.5	0.21
6	Unadjusted	13.8 ± 3.1	14.7 ± 1.2	**0.00**
	Adjusted	14 ± 3	14.9 ± 1.5	**0.00**
Floor mat (jumps)				
4	Unadjusted	4.4 ± 0.7	4.9 ± 0.3	**0.00**
	Adjusted	4.3 ± 0.8	4.8 ± 0.2	**0.00**
5	Unadjusted	4.7 ± 0.6	4.9 ± 0.2	**0.02**
	Adjusted	4.8 ± 0.5	4.9 ± 0.4	**0.01**
6	Unadjusted	4.7 ± 3.1	4.9 ± 0.3	**0.00**
	Adjusted	4.8 ± 3.3	4.9 ± 0.5	**0.00**

The values in bold indicate a statistical significance for *p* < 0.05.

**Table 4 ijerph-17-00797-t004:** Linear regression model for fitness level and gross motor skills outcomes adjusted by gender, age in months, height, weight, and BMI.

Variable	Gender	R^2^	Estimate	SE	*p*
Long jump	Boys	62%	5.6	2.4	**0.02**
	Girls	56%	8.9	1.4	**0.00**
Right dinamometry	Boys	73%	0.59	0.18	**0.00**
	Girls	71%	0.46	0.15	**0.00**
Left dinamometry	Boys	54%	1.2	0.21	**0.00**
	Girls	58%	0.45	0.15	**0.00**
Dominant hand dinamometry	Boys	75%	0.36	0.19	**0.02**
	Girls	75%	0.23	0.11	**0.00**
No dominant hand dinamometry	Boys	66%	0.31	0.15	**0.01**
	Girls	67%	0.27	0.15	**0.00**
Sit and reach	Boys	56%	1.3	0.22	**0.05**
	Girls	57%	0.78	0.42	**0.08**
4 × 10 m	Boys	73%	0.13	0.09	**0.05**
	Girls	71%	0.56	0.14	**0.00**
20-m shuttle run test	Boys	88%	0.15	0.08	**0.05**
	Girls	87%	0.35	0.06	**0.00**
Estimate VO2max	Boys	86%	0.22	0.18	**0.00**
	Girls	86%	0.67	0.14	**0.00**
Catching	Boys	59%	0.41	0.23	**0.02**
	Girls	61%	1.3	0.2	**0.00**
Aiming	Boys	58%	0.21	0.11	**0.00**
	Girls	66%	0.4	0.26	**0.01**
Right balance	Boys	70%	1.6	1	**0.02**
	Girls	68%	0.36	0.16	**0.00**
Leght balance	Boys	71%	0.43	0.23	**0.00**
	Girls	74%	0.47	0.25	**0.05**
Balance dominant leg	Boys	65%	0.45	0.12	**0.00**
	Girls	61%	0.39	0.15	**0.00**
Balance no dominant leg	Boys	67%	0.33	0.23	**0.04**
	Girls	67%	0.31	0.24	**0.05**
Tip-toes	Boys	59%	0.35	0.12	**0.05**
	Girls	62%	0.68	0.22	**0.03**
Floor mat	Boys	66%	0.3	0.1	**0.05**
	Girls	63%	0.02	0.01	**0.00**

The values in bold indicate a statistical significance for *p* < 0.05.

**Table 5 ijerph-17-00797-t005:** Percentile values for girls in fitness level.

Age (Years)	Long Jump (cm)	Speed/agility ^¥^ (Sec)	Right Dynamometry (kg)	Left Dynamometry (kg)	Dominant Hand Dynamometry (kg)	No Dominant Hand Dynamometry (kg)	Flexibility (cm)	20-m Shuttle Run Test (Stage) ‡	Estimated VO2max (mL/kg/min)
**4 (*n* = 92)**									
**p10**	42	23.5	2	2	2.6	2.3	23	0.4	46
**p20**	49.2	20.9	2.2	2.2	3.2	2.5	25.6	1	46
**p30**	54	19.4	2.2	5	3.4	2.5	27	1	46
**p40**	63	18.9	6	5.2	6	6	29	1.1	46.1
**p50**	67.5	18.6	6.1	5.5	6.7	6.6	29	1.2	46.9
**p60**	71	18.2	6.3	5.8	7.4	6.8	30	1.3	46.9
**p70**	78	17.1	6.8	6.2	8.3	7.4	31	1.4	46.9
**p80**	80	16.6	7.2	6.8	8.9	7.8	32	2	49
**p90**	90	16.3	7.4	7.9	9.3	8	34	2.2	49
**p100**	104	16.1	7.8	8.7	9.9	8.4	41	3.2	53.1
**5 (*n* = 126)**									
**p10**	60	25.1	5	5.1	5.2	5.2	26.4	0.3	46
**p20**	69	19.9	5.2	5.6	5.7	5.4	30	1	46
**p30**	72.1	18.7	5.6	5.8	6.3	5.7	31	1.1	46
**p40**	75.8	18.1	6.2	6.2	6.7	6.4	32	1.1	46.9
**p50**	78	17.8	6.5	6.9	7.4	6.7	33	1.2	46.9
**p60**	82	17.4	6.8	7.4	8.2	7.2	34	1.3	46.9
**p70**	86	16.4	7.3	7.7	9.1	7.8	35	1.9	48.7
**p80**	92	16.4	8.5	8.2	9.6	8.6	36	2	49
**p90**	100.6	15.8	9.1	8.9	10.4	9.3	37.6	2.3	49
**p100**	109	14.7	9.4	9.9	11	10	43	3.2	51.1
**6 (*n* = 178)**									
**p10**	66	20.4	5.8	5.5	5.8	5.8	25	1.1	46
**p20**	72	18.2	5.9	6.3	6.2	5.9	28	1.3	46.9
**p30**	77	17.5	6.3	6.8	7	6.4	29	1.4	46.9
**p40**	81	16.9	7.2	7.1	7.6	7.3	31.5	2	49
**p50**	85	16.3	7.8	7.8	8.2	7.8	33	2.1	49
**p60**	90	15.9	9	8.1	9	8.9	35	2.1	49
**p70**	95	15.8	9.4	8.7	9.6	9.5	36	2.2	49
**p80**	101	15.6	9.9	9.7	10.3	10.1	36	2.4	49
**p90**	108	15.2	10.3	11.2	13.4	12.6	38	3.4	51.1
**p100**	140	14.8	14.1	15	15.7	15	45	4.4	53.1

**^¥^** Less time (in sec) indicates better fitness level, ‡ 1 stage = 1 min.

**Table 6 ijerph-17-00797-t006:** Percentile values for girls in gross motor skills.

Age (Years)	Catching (Numbers)	Aiming (Numbers)	Right Balance (Sec)	Left Balance (Sec)	Dominant Leg Balance (Sec)	No Dominant Leg Balance (Sec)	Tip-Toes (Steps/Numbers)	Floor Mat (Jumps)
**4 (*n* = 92)**								
**p10**	2	1	3	3	5	3	8	0
**p20**	3	2	4.6	5.6	7	5	14.2	1
**p30**	4	3	6	6	8	7	15	1
**p40**	6	3	10	10	14	12	15	2
**p50**	6	4	15.5	15.5	16	18	15	3
**p60**	7	4	21	21	22	21	15	4
**p70**	9	5	24	24	28	24	15	4
**p80**	9	6	26	26	30	27	15	5
**p90**	10	9	30	30	30	30	15	5
**p100**	10	10	30	30	30	30	15	5
**5 (*n* = 126)**								
**p10**	4.7	1	6.7	6	8	6	5	1
**p20**	5	2	8	8	12	8	15	1
**p30**	7	2	10	11	17	14	15	2
**p40**	8	3	11.2	14	21	19	15	3
**p50**	9	3	14	20	24	22	15	3
**p60**	9	4	16.2	25.2	27	26	15	4
**p70**	10	5	24.7	30	30	28	15	5
**p80**	10	6	30	30	30	30	15	5
**p90**	10	7	30	30	30	30	15	5
**p100**	10	9	30	30	30	30	15	5
**6 (*n* = 178)**								
**p10**	7	2	12	10	15	9	15	1
**p20**	8	3	17	15	20	12	15	2
**p30**	9	4	25	19	26	20	15	3
**p40**	10	4	30	26	30	30	15	3
**p50**	10	5	30	30	30	30	15	4
**p60**	10	5	30	30	30	30	15	5
**p70**	10	6	30	30	30	30	15	5
**p80**	10	7	30	30	30	30	15	5
**p90**	10	8	30	30	30	30	15	5
**p100**	10	10	30	30	30	30	15	5

**Table 7 ijerph-17-00797-t007:** Percentile values for boys in fitness level.

Age (Years)	Long Jump (cm)	Speed/Agility ^¥^ (Sec)	Right Dynamometry (kg)	Left Dynamometry (kg)	Dominant Hand Dynamometry (kg)	No Dominant Hand Dynamometry (kg)	Flexibility (cm)	20-m Shuttle Run Test (stage) ‡	Estimated VO2max (mL/kg/min)
**4 (*n* = 54)**									
**p10**	32	21	5	5	6	4.8	23	0.4	46
**p20**	58	19.4	5.2	5	6.6	5.3	24	1	46
**p30**	69	18.8	5.3	5.6	7	5.9	26	1	46
**p40**	78	18.4	5.5	5.8	7.3	6.4	26	1	46.9
**p50**	83	17.8	5.6	6	7.8	6.6	27	1.1	46.9
**p60**	90	17.2	6.6	6.6	8.2	6.9	28	1.3	46.9
**p70**	93	16.5	7.2	6.8	8.5	7.1	28	1.4	46.9
**p80**	95	16.3	7.4	7.5	8.9	7.5	30	2	49
**p90**	97	16.2	8.4	7.8	9.1	8.1	32	2	49
**p100**	111	15.4	8.8	8.7	9.5	8.8	34	3.2	51.1
**5 (*n* = 112)**									
**p10**	62	27.9	5.3	5.2	5.7	5.5	26.5	1	46
**p20**	74.2	18.5	6.5	5.5	6.7	5.8	27	1.1	46
**p30**	80	18.2	7.1	6	7.3	6.4	29	1.3	46.9
**p40**	88	17.3	7.7	6.6	7.7	6.9	30	1.4	46.9
**p50**	91	16.8	7.9	6.8	8.3	7.2	31	2	49
**p60**	97	16.5	8.5	7.7	9	7.7	32	2	49
**p70**	100	15.9	9.1	7.8	9.8	8.2	34	2.2	49
**p80**	106	15.7	9.6	9.1	10.4	8.9	34	2.3	49
**p90**	116	15.3	10	9.9	10.9	9.4	36	2.5	49
**p100**	140	14.4	10.6	11.2	12.1	10.2	41	3	51.1
**6 (*n* = 166)**									
**p10**	72.1	25.2	6.6	6.2	6.8	6.7	23	1	46
**p20**	80	18.2	7.4	7.2	7.5	7.3	27	1.1	46
**p30**	85	17.2	8.1	8.2	8.3	7.9	28	1.3	46.9
**p40**	91.8	16.7	8.5	8.6	8.9	8.4	29	1.4	46.9
**p50**	97	16.4	8.9	9.4	9.5	9	30	1.5	46.9
**p60**	101.4	16.2	9.6	9.6	10	9.4	31	2.1	49
**p70**	105.9	15.7	10.3	9.9	10.4	9.9	32	2.2	49
**p80**	110	15.4	10.6	10.6	11.1	10	34	2.4	49
**p90**	126.3	14.9	11.4	12.1	14.8	12	36	3.2	51.1
**p100**	220	14.3	14.7	16.7	17	14.5	42	4.2	53.1

**^¥^** Less time (in sec) indicates better fitness level, ‡ 1 stage = 1 min.

**Table 8 ijerph-17-00797-t008:** Percentile values for boys in gross motor skills.

Age (Years)	Catching (Number)	Aiming (Number)	Right Balance (Sec)	Left Balance (Sec)	Dominant Leg Balance (Sec)	No Dominant Leg Balance (Sec)	Tip-Toes (Steps/Numbers)	Floor Mat (Jumps)
**4 (*n* = 54)**								
**p10**	3	1	2	3	4	2	8	0
**p20**	5	2	5	5	7	5	14.2	1
**p30**	6	2.5	6	6	10	9	15	1
**p40**	7	3	7	8	14	12	15	1
**p50**	8	3	8	11	17	15	15	2
**p60**	9	4	13	13	23	19	15	3
**p70**	9	4.5	15.5	20	27	22	15	3
**p80**	10	7	16	21.2	30	24	15	4
**p90**	10	8	20	24	30	27	15	5
**p100**	10	10	21	30	30	30	15	5
**5 (*n* = 112)**								
**p10**	5	2	5	3	5	4	6	1
**p20**	7	3	8	6	9	7	13	2
**p30**	8	4	12	8	14	11	15	3
**p40**	9	4	15	13	20	16	15	3
**p50**	10	4	17	17	24	21	15	4
**p60**	10	5	26	20	28	25	15	4
**p70**	10	6	30	30	30	28	15	4
**p80**	10	6	30	30	30	30	15	5
**p90**	10	7	30	30	30	30	15	5
**p100**	10	9	30	30	30	30	15	5
**6 (*n* = 166)**								
**p10**	6.7	2	3.7	5	7	4	10	1
**p20**	9	3	7	9	14	8	15	2
**p30**	9	4	9.3	13	19	13	15	2
**p40**	10	4	15	16.8	23	17	15	3
**p50**	10	5	20	20.1	27	21	15	4
**p60**	10	6	23.2	30	30	25	15	5
**p70**	10	7	30	30	30	29	15	5
**p80**	10	7	30	30	30	30	15	5
**p90**	10	8	30	30	30	30	15	5
**p100**	10	10	30	30	30	30	15	5

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
