# Peer review of "Reference Values for Fitness Level and Gross Motor Skills of 4–6-Year-Old Chilean Children"

_ijerph, 2020, doi:10.3390/ijerph17030797_

Round 1

Reviewer 1 Report

The issue discussed by the researchers is  important for practice, even though it is already explored in specialist literature, including authors’ national literature according to the references. Authors focus on selected indicators of preschool children’s physical development. Detailed comments on the research intent, study implementation, and theoretical aspects are presented below:

In the theoretical introduction, authors should emphasize that fitness level and cardiorespiratory fitness have both innate (anatomical and physiological) and environmental foundations in the form of specific health-related behaviors and lifestyle. This is important here due to the research intent and its practical implications, i.e. the development of standards for specialists working with children to increase the development of children's physical competencies.

The authors write (p.2, 45-47): "Considering (...) it is necessary to include tests to assess the fitness level of children in the school settings." Please provide a more thorough justification for the need for such tests in the context of the data collected so far regarding the population of children living in Chile / South America.

It is doubtful whether the reference values developed on the basis of a study involving children living in only one city are representative of the entire Chilean population. For example, we do not know how far the children living in this city deviate in terms of BMI and other analyzed characteristics (weight, height) from the entire population living in this country. The authors are aware of this limitation of their research. Perhaps it is worth giving up this aspect of analysis or clearly indicate that the established reference values apply to children from a specific region, and subsequent nationwide studies will provide more data to verify that.

There is no information as to whether the Movement Assessment Battery for Children is adapted for the needs of research and what the results of this adaptation are (e.g. reliability indicators).

Author Response

Reviewer 1:

The issue discussed by the researchers is important for practice, even though it is already explored in specialist literature, including authors’ national literature according to the references. Authors focus on selected indicators of preschool children’s physical development. Detailed comments on the research intent, study implementation, and theoretical aspects are presented below:

Comments 1: In the theoretical introduction, authors should emphasize that fitness level and cardiorespiratory fitness have both innate (anatomical and physiological) and environmental foundations in the form of specific health-related behaviors and lifestyle. This is important here due to the research intent and its practical implications, i.e. the development of standards for specialists working with children to increase the development of children's physical competencies.

Authors: Thank you for the reviewer’s comment. As suggested, we have included a new sentence in introduction section as follows:

Lines 55 to 57.

“It should be highlighted that both, fitness level and cardiorespiratory fitness, depend on genetic factors [8], including anatomical and physiological, and environmental factors [9], such as behaviour and lifestyles aspects.”

Comments 2: The authors write (p.2, 45-47): "Considering (...) it is necessary to include tests to assess the fitness level of children in the school settings." Please provide a more thorough justification for the need for such tests in the context of the data collected so far regarding the population of children living in Chile / South America.

Authors: Thank you for the reviewer’s comment. As suggested, we have included a justification for the need for these tests.

Lines 64 to 66.

… even more taking into account that Chile lacks records on indicators of fitness level [15], presenting a considerable gap in the literature for children 12 years old or younger [16].

Comments 3: It is doubtful whether the reference values developed on the basis of a study involving children living in only one city are representative of the entire Chilean population. For example, we do not know how far the children living in this city deviate in terms of BMI and other analyzed characteristics (weight, height) from the entire population living in this country. The authors are aware of this limitation of their research. Perhaps it is worth giving up this aspect of analysis or clearly indicate that the established reference values apply to children from a specific region, and subsequent nationwide studies will provide more data to verify that.

Authors: Thank you for the reviewer’s comment. It has been indicated the children specific region in methods section. As suggested, a limitation in discussion have been added.

Discussion, lines 54 and 56.

“ii) because the sample is not probabilistic and comes from only one region of Chile, inferences to Chilean children should be made with caution. Therefore, more studies are needed at the national level that provide data to verify what is found here.

Comments 4: There is no information as to whether the Movement Assessment Battery for Children is adapted for the needs of research and what the results of this adaptation are (e.g. reliability indicators).

Authors: Thank you for the reviewer’s comment. Spanish version validated and reliability values have been included.

Lines 130 to 132.

“The validated Spanish version of the second edition of the Movement Assessment Battery for Children (MABC-2) [36], has proven to be an instrument with adequate psychometric properties of reliability (α de Cronbach >0.60; κ=1; CCI=0.85-0.99) [37]…”

Reviewer 2 Report

Authors undertake an important topic related to children and their broadly understood gross motor skills. This is really important since those skills are the basis for everyday activities at school, at home and in the community

The course navette test (this is physical literacy that no everybody must know). I think that short definition of each test should be given in a supplement

L26: Differences in maximum oxygen volume are significant for boys at 5 years, and for girls at 6 years old (what authors mean by those differences? Differences in oxygen level between two different age groups – that is incomparable. Authors probably mean oxygen level before and after physical activity

L72: students? Children or students performance was measured?

L100 Authors write about mean and SD (how many times the tests were performed?). From how many replication this statistics were measured?

Table 2 – index value near sec – please correct this

P significance below table 2 (p<0,05) there should be dot not comma

Discussion part – line 9: For VO2 max, there are no large differences by gender at 4 years, 9

whereas at 5 years, there are significant differences in favour of boys, and at 6 years in favour of girls. – please explain this more deeply

It would be valuable to provide some tips (guidelines) for the unification of the overall fitness among 4-5 yr old girls and boys

Author Response

Reviewer 2:

Authors undertake an important topic related to children and their broadly understood gross motor skills. This is really important since those skills are the basis for everyday activities at school, at home and in the community

Comments 1: The course navette test (this is physical literacy that no everybody must know). I think that short definition of each test should be given in a supplement

Authors: Thank you for the reviewer’s comment. As suggested, we have included in a supplementary file a brief definition of fitness tests.

Comments 2: L26: Differences in maximum oxygen volume are significant for boys at 5 years, and for girls at 6 years old (what authors mean by those differences? Differences in oxygen level between two different age groups – that is incomparable. Authors probably mean oxygen level before and after physical activity

Authors: Thank you for the reviewer’s comment. The sentence has been modified as follows:

Lines 33 to 35.

“For estimated maximum volume of oxygen, at 5 years old there are significant differences in favour of boys, while at 6 years old in favour of girls.”

Comments 3: L72: students? Children or students performance was measured?

Authors: Thank you for the reviewer’s comment. We have changed students for children.

Comments 4: L100 Authors write about mean and SD (how many times the tests were performed?). From how many replication this statistics were measured?

Authors: Thank you for the reviewer’s comment. As suggested we have included information about how many times the tests were performed and how the values used were obtained.

Lines 116 and 117.

“The children were encouraged to run as long as possible during the test, and the last half of the stage reached by the participants was recorded.”

Lines 118 to 120.

“… the children were asked to jump horizontally to reach the maximum distance, the result was recorded in cm as the best value of the three attempts made.”

Lines 121 and 122.

“… two attempts were made for each hand, the highest values were recorded in kg.”

Lines 122 to 124.

“… two attempts were made with a 5-min rest between them and the lowest value of the two was recorded in seconds.”

Lines 124 and 125.

            “…the result was recorded in cm as the best value of the two attempts made.”

Comments 5: Table 2 – index value near sec – please correct this

Authors: Thank you for the reviewer’s comment. Done

Comments 6: P significance below table 2 (p<0,05) there should be dot not comma

Authors: Thank you for the reviewer’s comment. Done.

Comments 7: Discussion part – line 9: For VO2 max, there are no large differences by gender at 4 years, 9

Authors: Thank you for the reviewer’s comment. The sentence has been modified as follows:

Discussion, lines 10 and 11.

“For estimated VO2 max, there are no significant differences by gender at 4 years,...”

Comments 8: whereas at 5 years, there are significant differences in favour of boys, and at 6 years in favour of girls. – please explain this more deeply

Authors: Thank you for the reviewer’s comment. As suggested, we have modified the sentence as follows:

Discussion, lines 13 to 17.

“…which could be because the maximum values of VO2 max are relative (ml/kg/min) and these differences could be explained by weight, however, at 6 years old the differences are minimal (0.1 kg), which could explain that the value in favour of girls is a consequence of increased physical activity that would benefit an increase in VO2 max.”

Comments 9: It would be valuable to provide some tips (guidelines) for the unification of the overall fitness among 4-5 yr old girls and boys

Authors: Thank you for the reviewer’s comment. As suggested, some recommendations have been provided.

Discussion, lines 48 to 51.

“Thus, it is recommended that from an early age both girls and boys are encouraged to perform in different types of physical activity, where there is no sex-based exclusion, which favours a similar development in physical condition and motor competence.”

Reviewer 3 Report

Major comments:

This cross-sectional study was to investigate and to show reference values of physical fitness and motor skills in 4-6-year old Chilean children. These results provide normative data for physical fitness and motor skills in 4-6-year-old Chilean boys and girls. These findings are of important clinical/educational significance, but the manuscript has some concerns.

The current study demonstrates the reference values of fitness and motor skills tests in each chronological year-age in Tables 3 to 5. However, in these ages of young children, there are large individual differences in growth and development even if children are the same month ages. In addition, body size growth (height and weight) influences on physical fitness levels. Therefore, the authors should show reference values of fitness and motor skills tests after correcting for body size and month ages. Please see the following previous studies that show normative data of physical fitness related to month-ages and body size status.

ref#1: Molenaar HM, Selles RW, Zuidam JM, Willemsen SP, Stam HJ, Hovius SE. Growth diagrams for grip strength in children. Clin Orthop Relat Res. 2010 Jan;468(1):217-23.

ref#2: Wind AE1, Takken T, Helders PJ, Engelbert RH. Is grip strength a predictor for total muscle strength in healthy children, adolescents, and young adults? Eur J Pediatr. 2010 Mar;169(3):281-7.

ref#3: Ploegmakers JJ, Hepping AM, Geertzen JH, Bulstra SK, Stevens M. Grip strength is strongly associated with height, weight, and gender in childhood: a cross sectional study of 2241 children and adolescents providing reference values. J Physiother. 2013 Dec;59(4):255-61.

The authors should illustrate a flow diagram of participant recruitment and exclusion with reasons. In addition, were there children with special needs or ADHD? Were the children included in the analysis?

Please add descriptions of the following points and procedures for fitness tests: How did the authors evaluate maximal effort or exhaustion of these ages of young children in 20-m shuttle run test? How the authors assess children’s understanding of rules or procedures for the test? If the authors have data for the reproducibility of physical fitness and motor skills tests used in the current study, the authors should demonstrate the data.

Why did the authors compare boys and girls? The reviewer thinks the comparison could be outside the main purpose of the current study. Please state rationales for examining gender/sex differences in these participants.

The dominant hand/leg shows higher values of muscular strength than a non-dominant hand/leg does. In the current manuscript, the authors showed only data from right and left hands, but did not present data of the dominant/non-dominant hands separately, thus please revise and clarify that.

Minor comments

Please revise and add rationales for the current study in the last paragraph of the Introduction (L62 to L64). As the authors stated, there may be not clear cut-off values of physical fitness and motor skills in Latin American countries. However, there are international established reference values. Why do the authors need to organize the reference values of fitness and motor skills at the present?

The authors show the values of VO2max, but the data are estimated values by calculating 20-m shuttle run test laps. If so, the authors should change from “VO2max” to “estimated VO2max.” In addition, please change from the term “the Course Navette” to the term “20-m shuttle run test” throughout the manuscript

In Table 2, the authors show data of fitness tests as “Total”, “Boys” and “Girls”. Please reconsider the necessity for presenting data of “Total.”

I hope these comments will be helpful.

Author Response

Reviewer 3.

Major comments:

This cross-sectional study was to investigate and to show reference values of physical fitness and motor skills in 4-6-year old Chilean children. These results provide normative data for physical fitness and motor skills in 4-6-year-old Chilean boys and girls. These findings are of important clinical/educational significance, but the manuscript has some concerns.

Comments 1: The current study demonstrates the reference values of fitness and motor skills tests in each chronological year-age in Tables 3 to 5. However, in these ages of young children, there are large individual differences in growth and development even if children are the same month ages. In addition, body size growth (height and weight) influences on physical fitness levels. Therefore, the authors should show reference values of fitness and motor skills tests after correcting for body size and month ages. Please see the following previous studies that show normative data of physical fitness related to month-ages and body size status.

Authors: Thank you for the reviewer’s comment. As suggested, we have performed linear regression analysis, which can be seen in Table 4. Additionally, we have included information in methods and results sections.

Lines 145 to 147

“In order to establish the influence of gender, age, height, weight, and body mass index (BMI) on fitness level and gross motor skills in more detail, we performed a linear regression adding them as fixed factors [38-40].”

Lines 182 and 183

“Age, height, weight, and BMI are strongly associated with fitness level and gross motor skills in both, boys and girls (Table 4).”

Table 4. Linear regression model for fitness level and gross motor skills outcomes adjusted by gender, age, height, weight, and BMI.

Variable

Gender

R2

Estimate

SE

p

Long jump

Boys

62%

5.6

2.4

0.02

Girls

56%

8.9

1.4

0.00

Right dinamometry

Boys

73%

0.59

0.18

0.00

Girls

71%

0.46

0.15

0.00

Left dinamometry

Boys

54%

1.2

0.21

0.00

Girls

58%

0.45

0.15

0.00

Dominant hand dinamometry

Boys

75%

0.36

0.19

0.02

Girls

75%

0.23

0.11

0.00

No dominant hand dinamometry

Boys

66%

0.31

0.15

0.01

Girls

67%

0.27

0.15

0.00

Sit and reach

Boys

56%

1.30

0.22

0.05

Girls

57%

0.78

0.42

0.08

4x10m

Boys

73%

0.13

0.09

0.05

Girls

71%

0.56

0.14

0.00

20-m shuttle run test

Boys

88%

0.15

0.08

0.05

Girls

87%

0.35

0.06

0.00

Estimate Vo2 max

Boys

86%

0.22

0.18

0.00

Girls

86%

0.67

0.14

0.00

Catching

Boys

59%

0.41

0.23

0.02

Girls

61%

1.30

0.20

0.00

Aiming

Boys

58%

0.21

0.11

0.00

Girls

66%

0.40

0.26

0.01

Right balance

Boys

70%

1.60

1.00

0.02

Girls

68%

0.36

0.16

0.00

Leght balance

Boys

71%

0.43

0.23

0.00

Girls

74%

0.47

0.25

0.05

Balance dominant leg

Boys

65%

0.45

0.12

0.00

Girls

61%

0.39

0.15

0.00

Balance no dominant leg

Boys

67%

0.33

0.23

0.04

Girls

67%

0.31

0.24

0.05

Tip-toes

Boys

59%

0.35

0.12

0.05

Girls

62%

0.68

0.22

0.03

Floor mat

Boys

66%

0.30

0.10

0.05

Girls

63%

0.02

0.01

0.00

The values in bold indicate a statistical significance for p < 0.05

Comments 2: The authors should illustrate a flow diagram of participant recruitment and exclusion with reasons. In addition, were there children with special needs or ADHD? Were the children included in the analysis?

Authors: Thank you for the reviewer’s comment. As suggested we have added flow diagram of participant recruitment and exclusion with reasons. Only children with autism spectrum disorders were excluded. Children with special needs or ADHD were included in the analysis.

Comments 3: Please add descriptions of the following points and procedures for fitness tests: How did the authors evaluate maximal effort or exhaustion of these ages of young children in 20-m shuttle run test? How the authors assess children’s understanding of rules or procedures for the test? If the authors have data for the reproducibility of physical fitness and motor skills tests used in the current study, the authors should demonstrate the data.

Authors: Thank you for the reviewer’s comment. As suggested, we have added the following information:

Lines 116 and 117.

“The children were encouraged to run as long as possible during the test, and the last half of the stage reached by the participants was recorded.”

Lines 105 and 106.

“These graduates were responsible for ensuring sure that the instructions and procedures were understood in each test”.

Lines 130 to 132.

“The validated Spanish version of the second edition of the Movement Assessment Battery for Children (MABC-2)[36], has proven to be an instrument with adequate psychometric properties of reliability (α de Cronbach >0.60; κ=1; CCI=0.85-0.99) [37]”.

Comments 4: Why did the authors compare boys and girls? The reviewer thinks the comparison could be outside the main purpose of the current study. Please state rationales for examining gender/sex differences in these participants.

Authors: Thank you for the reviewer’s comment. As suggested, we have clarified this point as follows:

Lines 81 and 82.

“and that these differ by sex because at an early age, the evidence has demonstrated that there is a significant difference in fitness level and gross motor skills between boys and girls [23-26].”

Comments 5: The dominant hand/leg shows higher values of muscular strength than a non-dominant hand/leg does. In the current manuscript, the authors showed only data from right and left hands, but did not present data of the dominant/non-dominant hands separately, thus please revise and clarify that.

Authors: Thank you for the reviewer’s comment. In tables 3 and 4 data of dominant and non-dominant hands and legs were added. Additionally, this information has been included in the results section as follows:

Lines 160 to 162

“…right and left dynamometry (p from <0.00 to 0.12), dominant hand dynamometry (p from <0.00 to 0.00), no dominant hand dynamometry (p from <0.00 to 0.00).”

Minor comments

Comments 6: Please revise and add rationales for the current study in the last paragraph of the Introduction (L62 to L64). As the authors stated, there may be not clear cut-off values of physical fitness and motor skills in Latin American countries. However, there are international established reference values. Why do the authors need to organize the reference values of fitness and motor skills at the present?

 Authors: Thank you for the reviewer’s comment. As suggested, we have added the following information:

Lines 86 to 88.

“Which makes it necessary to have their own values, due to sociodemographic factors, the environment and the adequate nutrition of this population, which leads them to have different characteristics.”

Comments 7: The authors show the values of VO2max, but the data are estimated values by calculating 20-m shuttle run test laps. If so, the authors should change from “VO2max” to “estimated VO2max.” In addition, please change from the term “the Course Navette” to the term “20-m shuttle run test” throughout the manuscript

Authors: Thank you for the reviewer’s comment. Done.

Comments 8: In Table 2, the authors show data of fitness tests as “Total”, “Boys” and “Girls”. Please reconsider the necessity for presenting data of “Total.”

Authors: Thank you for the reviewer’s comment. Done.

Round 2

Reviewer 3 Report

Major comments:

Your revised manuscript was much improved. But, the revised manuscript has still minor concerns.

The current study demonstrates the reference values of fitness and motor skills tests in each chronological year-age in Tables 3 to 5. However, in these ages of young children, there are large individual differences in growth and development even if children are the same month ages. In addition, body size growth (height and weight) influences on physical fitness levels. Therefore, the authors should show reference values of fitness and motor skills tests after correcting for body size and month ages. Please see the following previous studies that show normative data of physical fitness related to month-ages and body size status.

Authors: Thank you for the reviewer’s comment. As suggested, we have performed a linear regression analysis, which can be seen in Table 4. Additionally, we have included information in the methods and results sections.

Comments to authors:

The authors did not understand my previous comment. The reviewer recommended the authors to adjust for body size and/or month ages as growth levels because reference values of physical fitness and gross motor skills in the manuscript would be influenced by body size and growth levels. But the authors added Table 4. Once again, the reviewer recommends that the authors should show reference values of physical fitness and gross motor skills with or without adjusting for these confounders.

Comments 2: The authors should illustrate a flow diagram of participant recruitment and exclusion with reasons. In addition, were there children with special needs or ADHD? Were the children included in the analysis?

Authors: Thank you for the reviewer’s comment. As suggested we have added a flow diagram of participant recruitment and exclusion with reasons. Only children with autism spectrum disorders were excluded. Children with special needs or ADHD were included in the analysis.

Reviewer comment: The authors addressed.

Comments 3: Please add descriptions of the following points and procedures for fitness tests: How did the authors evaluate maximal effort or exhaustion of these ages of young children in 20-m shuttle run test? How the authors assess children’s understanding of rules or procedures for the test? If the authors have data for the reproducibility of physical fitness and motor skills tests used in the current study, the authors should demonstrate the data.

Authors: Thank you for the reviewer’s comment. As suggested, we have added the following information:

Lines 116 and 117.

“The children were encouraged to run as long as possible during the test, and the last half of the stage reached by the participants was recorded.”

Lines 105 and 106.

“These graduates were responsible for ensuring sure that the instructions and procedures were understood in each test”.

Lines 130 to 132.

“The validated Spanish version of the second edition of the Movement Assessment Battery for Children (MABC-2)[36], has proven to be an instrument with adequate psychometric properties of reliability (α de Cronbach >0.60; κ=1; CCI=0.85-0.99) [37]”.

Reviewer comment: The author addressed.

Comments 4: Why did the authors compare boys and girls? The reviewer thinks the comparison could be outside the main purpose of the current study. Please state rationales for examining gender/sex differences in these participants.

Authors: Thank you for the reviewer’s comment. As suggested, we have clarified this point as follows:

Lines 81 and 82.

“and that these differ by sex because at an early age, the evidence has demonstrated that there is a significant difference in fitness level and gross motor skills between boys and girls [23-26].”

Reviewer comment: OK.

Comments 5: The dominant hand/leg shows higher values of muscular strength than a non-dominant hand/leg does. In the current manuscript, the authors showed only data from right and left hands, but did not present data of the dominant/non-dominant hands separately, thus please revise and clarify that.

Authors: Thank you for the reviewer’s comment. In tables 3 and 4 data of dominant and non-dominant hands and legs were added. Additionally, this information has been included in the results section as follows:

Lines 160 to 162

“…right and left dynamometry (p from <0.00 to 0.12), dominant hand dynamometry (p from <0.00 to 0.00), no dominant hand dynamometry (p from <0.00 to 0.00).”

Reviewer comment: The authors addressed.

Minor comments

Comments 6: Please revise and add rationales for the current study in the last paragraph of the Introduction (L62 to L64). As the authors stated, there may be not clear cut-off values of physical fitness and motor skills in Latin American countries. However, there are international established reference values. Why do the authors need to organize the reference values of fitness and motor skills at the present?

 Authors: Thank you for the reviewer’s comment. As suggested, we have added the following information:

Lines 86 to 88.

“Which makes it necessary to have their own values, due to sociodemographic factors, the environment and the adequate nutrition of this population, which leads them to have different characteristics.”

Reviewer comment: OK.

Comments 7: The authors show the values of VO2max, but the data are estimated values by calculating 20-m shuttle run test laps. If so, the authors should change from “VO2max” to “estimated VO2max.” In addition, please change from the term “the Course Navette” to the term “20-m shuttle run test” throughout the manuscript

Authors: Thank you for the reviewer’s comment. Done.

Reviewer comment: The authors addressed.

Minor comment:

p2, L68. the sentence is incomplete. Please revise.

Author Response

Reviewer 3

Your revised manuscript was much improved. But, the revised manuscript has still minor concerns.

The current study demonstrates the reference values of fitness and motor skills tests in each chronological year-age in Tables 3 to 5. However, in these ages of young children, there are large individual differences in growth and development even if children are the same month ages. In addition, body size growth (height and weight) influences on physical fitness levels. Therefore, the authors should show reference values of fitness and motor skills tests after correcting for body size and month ages. Please see the following previous studies that show normative data of physical fitness related to month-ages and body size status.

Authors:

We would like to thank you the thoughtful reviewer’s comment. As suggested, we have performed a linear regression analysis, which has been displayed in Table 4. Additionally, we have included information on this analysis in the methods and results sections as follow:

Methods: Lines125 to 129.

“In order to establish the influence of gender, age in months, height, weight, and body mass index (BMI) on fitness level and gross motor skills, a linear regression analysis was performed adding these variables as fixed factors [38-40].

ANCOVA model was used to assess differences in fitness level and gross motor skills, controlling for age in months and BMI, by gender.”

Comments 1: The authors did not understand my previous comment. The reviewer recommended the authors to adjust for body size and/or month ages as growth levels because reference values of physical fitness and gross motor skills in the manuscript would be influenced by body size and growth levels. But the authors added Table 4. Once again, the reviewer recommends that the authors should show reference values of physical fitness and gross motor skills with or without adjusting for these confounders.

Authors:

We would like to apologize for the inconvenience and thank again the reviewer’s comment. As suggested, we have performed analysis with the aim to show reference values of physical fitness and gross motor skills with and without adjusting for confounders. This information has been added in the tables of the paper.

Comments 2: p2, L68. the sentence is incomplete. Please revise.

Authors:

Thank you for the reviewer’s comment. Done.

Lines 64 to 68.

“These assessments require the existence of updated reference values that allow the categorization of individuals and groups according to fitness levels and gross motor skills. Additionally, these values might be established by sex as previous evidence has demonstrated significant differences in fitness levels and gross motor skills between boys and girls at early ages [23-26].”